# Cryo-EM reveals structural basis for human AIM/CD5L recognition of polymeric immunoglobulin M

Qu Chen [1,7], Kazuhiro Ishii [2,3,7], Haruka Mori[2], Akemi Nishijima[2], Satoko Arai [2] ✉, Toru Miyazaki [2,4,5] ✉ & Peter B. Rosenthal [6] ✉

Cell surface scavenger receptors contribute to homoeostasis and the response to pathogens and products associated with damage by binding to common molecular features on a wide range of targets. Apoptosis inhibitor of macrophage (AIM/CD5L) is a soluble protein belonging to the scavenger receptor cysteine-rich (SRCR) superfamily that contributes to prevention of a wide range of diseases associated with infection, inflammation, and cancer. AIM forms complexes with IgM pentamers which helps maintain high-levels of circulating AIM in serum for subsequent activation on release from the complex. The structural basis for AIM recognition of IgM as well as other binding targets is unknown. Here we apply cryogenic electron microscopy imaging (cryo-EM) to show how interfaces on both of AIM's C-terminal SRCR domains interact with the Fcμ constant region and J chain components of the IgM core. Both SRCR interfaces are also shown to contribute interactions important for AIM binding to damage-associated molecular patterns (DAMPs).

The Scavenger Receptor Cysteine-Rich (SRCR) domain is a conserved domain characterising a superfamily (SRCR-SF) of multi-domain proteins which play fundamental and diverse roles in homoeostasis and immune defence[1]. The -110 residue SRCR domains are predominantly classified by a pattern of either six cysteines (Group A) or eight cysteines (Group B)[2].

Apoptosis inhibitor of macrophage (AIM), also called CD5 antigen-like (CD5L), is a 36 kDa soluble protein that belongs to the SRCR superfamily Group B with three head-to-tail linked SRCR domains, mainly expressed by tissue-resident macrophages in the liver as well as lymphoid and inflamed tissues[3,4]. Initially characterised as a supporter of macrophage survival[5], AIM also contributes to the prevention of various diseases, including obesity[6–8], fatty liver disease[9], hepatocellular carcinoma[9,10], multiple sclerosis[11], fungus-induced peritonitis[12],

acute kidney injury[13,14], ocular hypertension[15] and is a promising drug target for cat kidney disease[16].

AIM is present in an IgM-bound form in serum, preventing it from renal excretion, which maintains high AIM levels in blood[14,17–19]. At the sites of damaged or dying cells caused by trauma or an infection, free AIM released from IgM can bind dead cell debris which contains damage-associated molecular patterns (DAMPs), thereby promoting the internalisation of dead cells by phagocytes[3,20,21]. Considering that the capacity of AIM to recognize DAMPs is prevented when associated with IgM[3], it is imperative to comprehensively elucidate the molecular mechanisms underlying AIM's recognition of both IgM and DAMPs. This understanding is pivotal for exploring the potential utilization of AIM protein as an innovative treatment for inflammatory conditions in various diseases, such as acute kidney injury (AKI)[13] and ischaemic

[1]Structural Biology Science Technology Platform, The Francis Crick Institute, London, UK. [2]The Institute for AIM Medicine, Tokyo, Japan. [3]Department of Physiological Chemistry and Metabolism, Graduate School of Medicine, The University of Tokyo, Tokyo, Japan. [4]LEAP, Japan Agency for Medical Research and Development, Tokyo, Japan. [5]Laboratoire d'ImmunoRhumatologie Moléculaire, Plateforme GENOMAX, Institut National de la Santé et de la Recherche Médicale UMR_S 1109, Faculté de Médecine, Fédération Hospitalo-Universitaire OMICARE, Fédération de Médecine Translationnelle de Strasbourg, Laboratory of Excellence TRANSPLANTEX, Université de Strasbourg, Strasbourg, France. [6]Structural Biology of Cells and Viruses Laboratory, The Francis Crick Institute, London, UK. [7]These authors contributed equally: Qu Chen, Kazuhiro Ishii. ✉e-mail: arai.satoko@iamaim.jp; tm@iamaim.jp; Peter.Rosenthal@crick.ac.uk

stroke[3]. 2D negative stain imaging of AIM/IgM showed that AIM sits in the groove of the asymmetric IgM pentamer[22], while the 3D structure of the complex at higher resolution is required to understand the association at the atomic level.

In this study, we determined the structural organisation of AIM/IgM by single particle cryo-EM, revealing the interactions between the two molecules and the mechanism by which AIM is stabilised in the serum by IgM. AIM uses interfaces containing a free cysteine and a metal binding site to bind to IgM and these interactions are further shown to be important for AIM binding to several DAMPs. These results, together with known structures and functional data for other SRCR domains, inform generally on ligand recognition by proteins in the SRCR-SF.

## Results

### Cryo-EM structure of human Fcμ/J/AIM

We expressed human pentameric IgM core which includes ten copies of Fcμ chains from Cμ2 to the C-terminal tailpiece and one J chain, as well as full-length human AIM (hAIM). The cryo-EM structure of human pentameric IgM core in complex with AIM is presented in Fig. 1a, b and Supplementary Figs. 1 and 2. As published previously, the density of the cryo-EM map of pentameric IgM core contains ten Cμ3 and Cμ4 domains (in grey) and a single J chain (in orange), with the absence of Cμ2 domains which are disordered in the map due to their flexible linkage. AIM engages pentameric IgM core with 1:1 stoichiometry. AIM (in cyan) occupies a groove of Fcμ/J, with SRCR2 and SRCR3 domains proximal to the J chain and subunit Fcμ5. The global map resolution is 3.6 Å, with local resolution at the interface predominantly ranging from 3 Å to 3.5 Å. This enabled us to identify side chain density

(Supplementary Figs. 3 and 4) and propose interactions. The density of the N-terminal SRCR1 domain appears only at lower contour level (Fig. 1c), indicating a flexible connection to SRCR2. The schematic diagram of the overall structure of human Fcμ/J/AIM is shown in Fig. 1d.

Comparison of the Fcμ/J/AIM to other pentameric IgM structures suggests that only minor changes in the IgM core occur on AIM binding, but nevertheless AIM binding has functional consequences for IgM. Polymeric immunoglobulin receptor (pIgR), a component of secretory IgM which mediates the IgM transcytosis through the mucosal epithelial cells, also occupies the cleft in the IgM pentamer when bound, and therefore pIgR and AIM cannot bind IgM at the same time (Supplementary Fig. 5a). This explains why AIM concentration in saliva and milk is significantly lower than that in blood[23]. It has been reported that AIM binding does not affect the high-specificity IgM Fc receptor (FcμR/TOSO/FAIM3) recognition of IgM[23]. Superposition of the Fcμ/J/AIM complex on the IgM/FcμR structure[24], shows that Fcμ/J/AIM can accommodate binding of all eight FcμR Ig domains without clashes (Supplementary Fig. 5b). Similarly, it has also been reported that AIM does not influence complement pathway activation[23], consistent with our Fcμ/J/AIM structure as the binding sites on IgM for C1q are at the tips of Cμ3 domains which are distant from AIM. However, the Fcα/μR (CD351) binding is affected by AIM[19,23] and this will require a high-resolution structure of Fcα/μR complexed with IgM to understand in detail.

### SRCR domains in AIM

hAIM contains three highly conserved cysteine-rich domains. As a group B SRCR protein, each SRCR domain contains a conserved

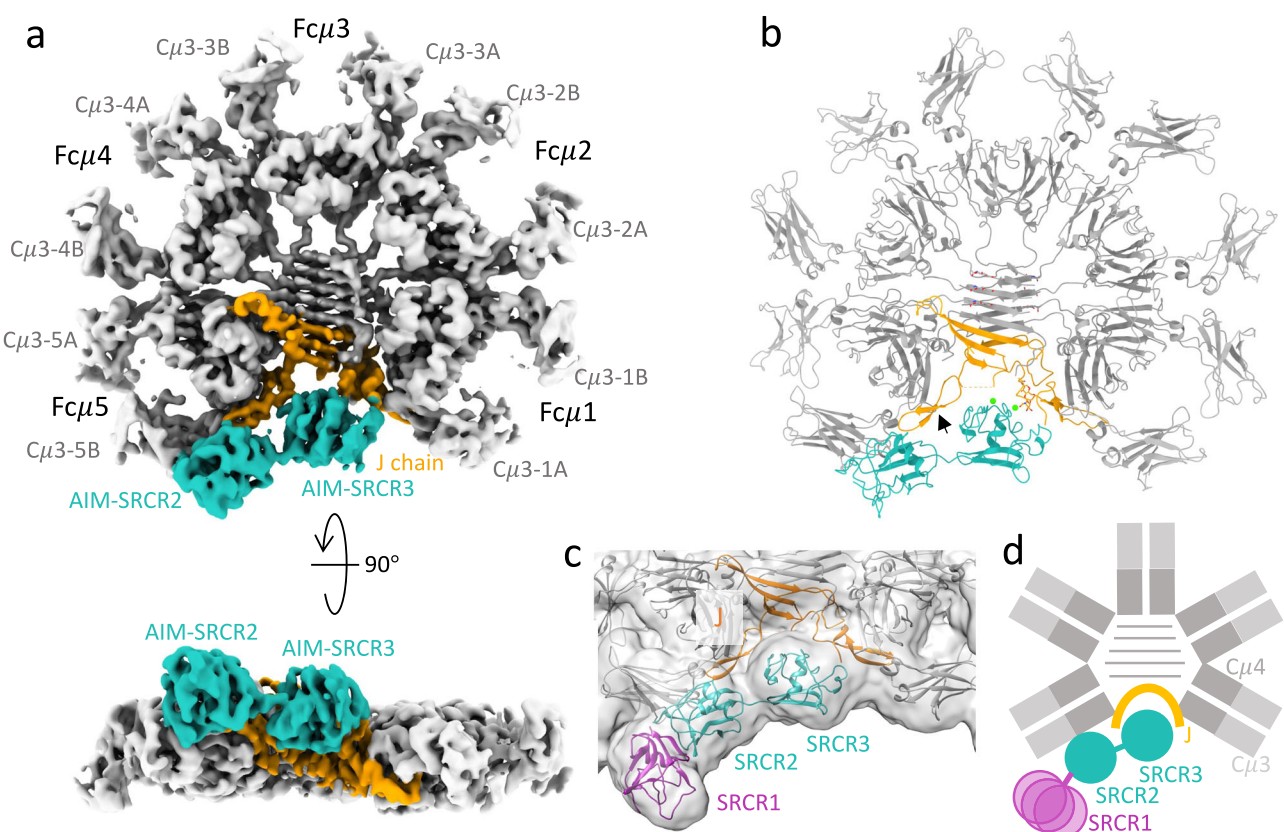

**Fig. 1 | Cryo-EM structure of Fcμ/J/AIM complex. a** Front (upper panel) and side (lower panel) view of the cryo-EM map of Fcμ/J/AIM complex. Fcμ chains are coloured in grey, J chain in orange, and AIM in cyan. The IgM subunits are numbered from Fcμ1 to Fcμ5 anticlockwise. **b** The model of Fcμ/J/AIM complex with the same colour codes as in (**a**). The black arrow points to the β-hairpin 2 of the J chain.

**c** Gaussian-filtered (Width 1.5) cryo-EM map at threshold showing density for the SRCR1 domain. Contour level t = 0.0353 (4σ). The Alphafold model of SRCR1 is shown in pink. **d** Schematic illustration of the overall structure of Fcμ/J/AIM complex.

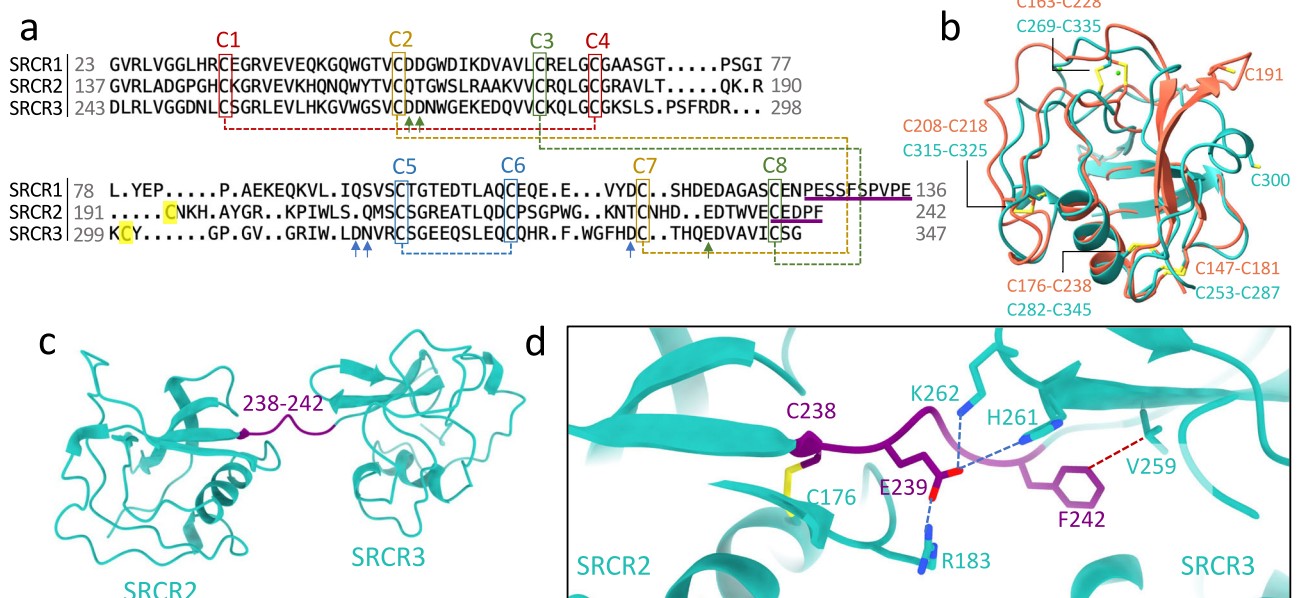

**Fig. 2 | SRCR domains of AIM. a** Structure-based sequence alignment of the three SRCR domains of AIM. SRCR2 and SRCR3 are cryo-EM structures and SRCR1 is the AF2 predicted model. The four conserved disulfide bonds are highlighted with dash lines with different colours. The unpaired cysteine in SRCR2 and SRCR3 are in yellow. The three green arrows indicate the residues of calcium binding site 1 in SRCR3 which are also conserved in SRCR1. The three blue arrows point at the calcium binding site 2 residues in SRCR3. The sequences of linkers between SRCR1 and SRCR2 and between SRCR2 and SRCR3 are underlined in purple. **b** Superposition of SRCR2 (red) and SRCR3 (cyan) cryo-EM models showing the four conserved disulfide bonds and the two unpaired cysteine residues (C191 and C300). **c** AIM model from the complex with IgM showing linker (highlighted in purple) between SRCR2 and SRCR3 domains. **d** Zoomed-in view of the linker-domain interactions with potential salt bridges as blue dashed lines and van der Waals contact as red dashed line.

pattern of four intradomain disulfide bonds (C1-C4, C2-C7, C3-C8, C5-C6, Fig. 2a, b) and highly similar folds. The root mean square deviation (RMSD) between SRCR2 and SRCR3 is 1.93 Å. The AlphaFold2 prediction for SRCR1 (O43866) is also highly similar to SRCR3 with an RMSD of 1.28 Å. An unpaired cysteine, which is exceptional in the SRCR-SF, is present in both SRCR2 (C191) and SRCR3 (C300) between C4 and C5 (Fig. 2a, b), but not in SRCR1. There are map densities for two ions at the SRCR3 interface with J chain (Supplementary Fig. 4) at canonical SRCR-SF metal ion binding sites[25,26]. The ions are likely to be calcium ions (Ca$^{2+}$) because Ca$^{2+}$ is required for AIM/IgM binding, and binding is reduced when other divalent cations such as Mg$^{2+}$ and Zn$^{2+}$ are present instead of calcium (Supplementary Fig. 6). Density corresponding to metal ions was not identified on SRCR2.

The linker between SRCR2 and SRCR3 is short, containing 5 residues (C238-F242, Fig. 2a, c). The beginning of the linker forms a disulfide bond with the main body of SRCR2 (C176-C238), which may be further stabilised by a salt bridge between R183 and E239 (Fig. 2d). The same residue E239 can interact with two adjacent polar or positively charged residues in SRCR3 (H261 and K262). The end of the linker (F242) makes a van der Waals contact to V259 in SRCR3 domain (Fig. 2d). These interactions contribute to the limited flexibility of the linkage between SRCR2 and SRCR3 in contrast to the flexible linker between SRCR1 and SRCR2 domains, which is twice as long, 10 residues, as shown in Fig. 2a.

**Recognition of IgM by AIM**

AIM recognises IgM at both Fcμ and J chain (Fig. 3a). The total buried surface area (BSA) between AIM and IgM is 994 Å$^2$, with 652 Å$^2$ between AIM and J chain and 342 Å$^2$ between AIM and subunit Fcμ5. The SRCR3 domain occupies the groove formed by β-hairpin 2 and 3 in J chain (Fig. 3a). Interestingly, β-hairpin 2 has been found to be disordered in IgM pentamer structures[24,27–30], but is stabilised by AIM due to being sandwiched between subunit Fcμ5 and the two SRCR domains. Between Fcμ5 and β-hairpin 2, Q81$^J$ can form a hydrogen bond with the carbonyl group of F358$^{Cμ3-5B}$, and several interactions occur between

residues T84-Q87 on J chain and N545-R550 on Cμ4-5B including a van der Waals contact between A85$^J$ to V547$^{Cμ4-5B}$ (Fig. 3b). This interaction is conserved in the human IgA dimer (A85$^J$-F443$^{Cα3}$)[31] as well as mouse IgA[32]. Other van der Waals interactions observed in IgA between Fcα and β-hairpin 2 are absent in IgM between Fcμ and β-hairpin 2[31], which could explain the higher flexibility of β-hairpin 2 in IgM in comparison to IgA (Supplementary Fig. 7a, b). Intriguingly, the AIM-stabilised β-hairpin 2 results in a shift in the orientation of the Cμ3-5B domain (Supplementary Fig. 7c, d). As previously described for J-chain interactions at Cμ3–1A[27], this potentially leads to an asymmetric conformation of the associated Fab, which may influence antigen recognition or complement binding.

SRCR2 domain is covalently coupled to Fcμ chain. The unpaired cysteine residue C191 in SRCR2$^{AIM}$ forms a disulfide bond with C414 in Cμ3-5B (Fig. 3c), which aligns with the fact that AIM does not dissociate from IgM during SDS–polyacrylamide gel electrophoresis in non-reducing conditions[22]. This disulfide bond is crucial, as the C191S mutant completely abolished the binding between IgM and AIM (Fig. 3e), as previously shown[22]. The corresponding cysteine in SRCR3, C300, is distant from the binding interface and remains unpaired (Fig. 3a). This free cysteine is not conserved in AIM in any other species, suggesting that it could potentially play a unique role in human AIM. In addition, K173 and R183 in SRCR2 form two salt bridges with β-hairpin 2 of J chain at E77 and E75 (Fig. 3c). SRCR3$^{AIM}$ sits between β-hairpin 2 and β-hairpin 3 of J chain and interacts with both loops. F329 and F332 on SRCR3 pack against P96 in β-hairpin 2 of J chain through a van der Waals contact, plus a salt bridge between D72 and R328 (Fig. 3c).

One of the two Ca$^{2+}$ (Ca$^{2+}$ site 1 at right in Fig. 3d) bridges D270, D271 and E339 on SRCR3$^{AIM}$, and the carbonyl group of N106 on the J chain. Interactions between SRCR3 and β-hairpin 3 of the J chain includes a hydrogen bond (D271/O$^{AIM}$-N106$^J$) and a salt bridge (D271$^{AIM}$-K107$^J$), as shown in Fig. 3d. Local cryo-EM map densities for all the interacting residues are presented in Supplementary Figs. 3 and 4. The second Ca$^{2+}$ (Ca$^{2+}$ site 2 at left in Fig. 3d) interacts with D311, N312, D334

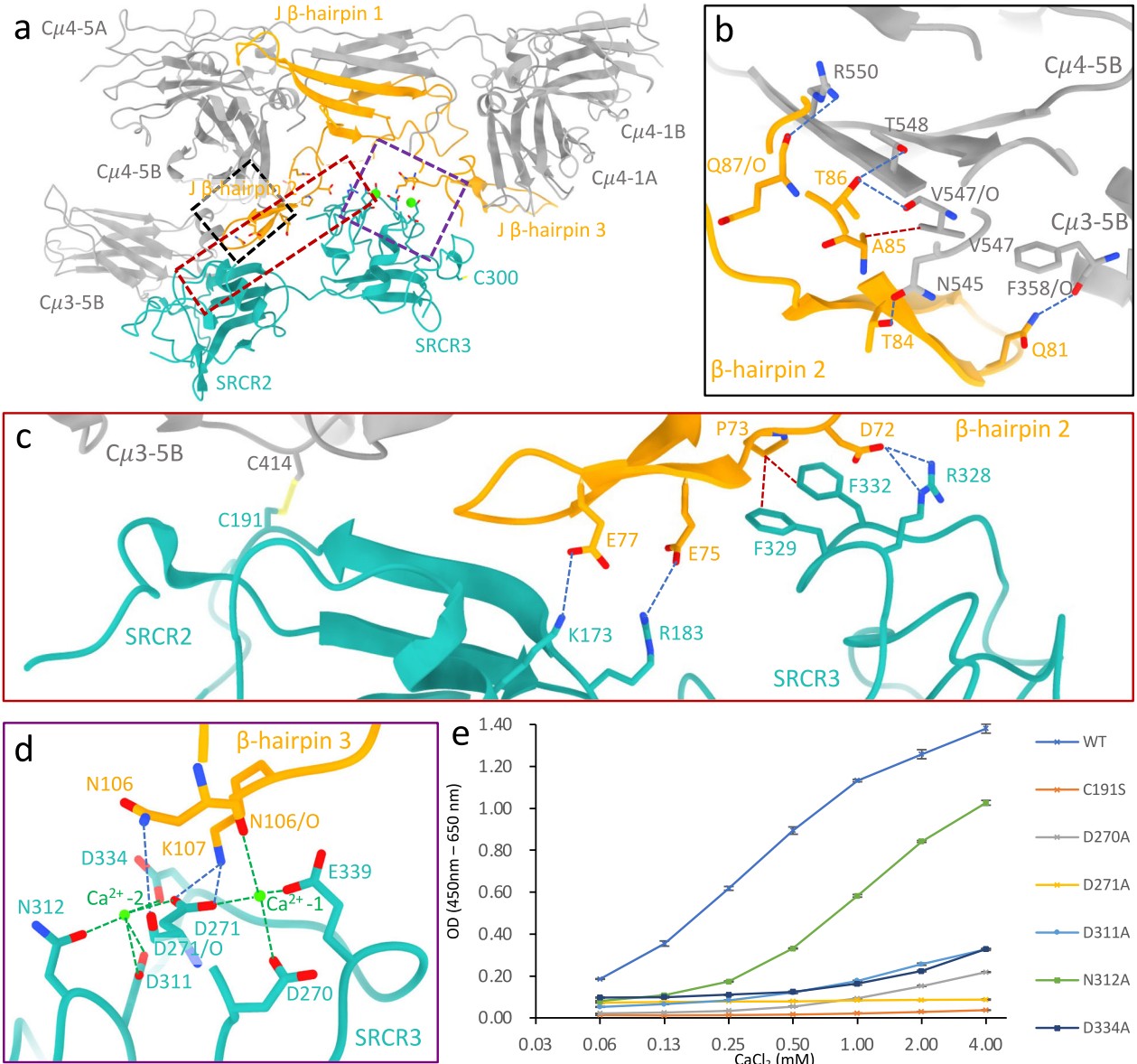

**Fig. 3 | Recognition of IgM by AIM. a** Model showing the Fcμ/J/AIM interfaces in the complex (lower region of model in Fig. 1b). **b** Interactions between the stabilised β-hairpin 2 and Cμ3 and Cμ4 domains in subunit Fcμ5 (black dashed box in (**a**)). **c** Disulfide bond, potential salt bridges (blue dashed lines) and van der Waals contact (red dashed lines) between the two SRCR domains in AIM and IgM (red dashed box in (**a**)). **d** Calcium binding sites (Ca²⁺-1 and Ca²⁺−2) on SRCR3 at the interface between AIM and J chain (purple dashed box in (**a**)). **e** Binding of AIM (WT and mutants) to IgM at different calcium ion concentrations in an ELISA-based assay. Error bars indicate the s.e.m. for three technical replicates. Data is representative of three biological replicates. Source data are provided as a Source Data file.

and D271 on SRCR3$^{AIM}$, but is not in position to directly interact with IgM at either Fcμ or J chain.

IgM/AIM binding is found to be Ca²⁺ concentration dependent, as shown in Fig. 3e; increasing the Ca²⁺ concentration promotes the binding of IgM and wild-type AIM. AIM mutants D270A or D271A reduce IgM binding even at increased Ca²⁺ concentration (Fig. 3e), which further supports their roles as ligands for the calcium that bridges AIM and IgM. Intriguingly, the mutants that disrupt Ca²⁺ site 2, including D311A, N312A and D334A, also significantly diminish binding to IgM (Fig. 3e), likely due to a local structure effect that destabilises Ca²⁺ site 1. N312A shows some degree of Ca²⁺ concentration-dependent binding, and its mutation may have the least effect on binding due to its distance from the first Ca²⁺ site. Interestingly, the pair of aspartic acid residues at Ca²⁺ site 1 is absent in SRCR2 but is conserved in SRCR1, indicated by the green arrows in Fig. 2a. However, mutating the two

aspartic acid residues (D50A, D51A) does not impact the binding between IgM and AIM (Supplementary Fig. 8a), which is expected as SRCR1 appears to be mobile when AIM binds to IgM.

To investigate whether the same interactions are used by AIM to recognise DAMPs, we performed binding assays for six DAMPs previously shown to be bound by AIM[3], with WT or mutated AIM (Fig. 4a–f). All the WT-AIM interactions with DAMPs exhibit Ca²⁺ concentration dependence, and mutating the two aspartic acid residues (D270A and D271A) abrogates the binding for all the cases, suggesting the indispensability of the first calcium site. As previously reported[3], the C191S mutant, on the other hand, diminished the binding between AIM and DAMPs to different levels (Fig. 4a–f), indicating that the disulfide bond contributes to but is not necessary for the association between AIM and DAMPs. The potential calcium binding site on SRCR1 is not involved in DAMPs binding (Supplementary Fig. 8b, c). These

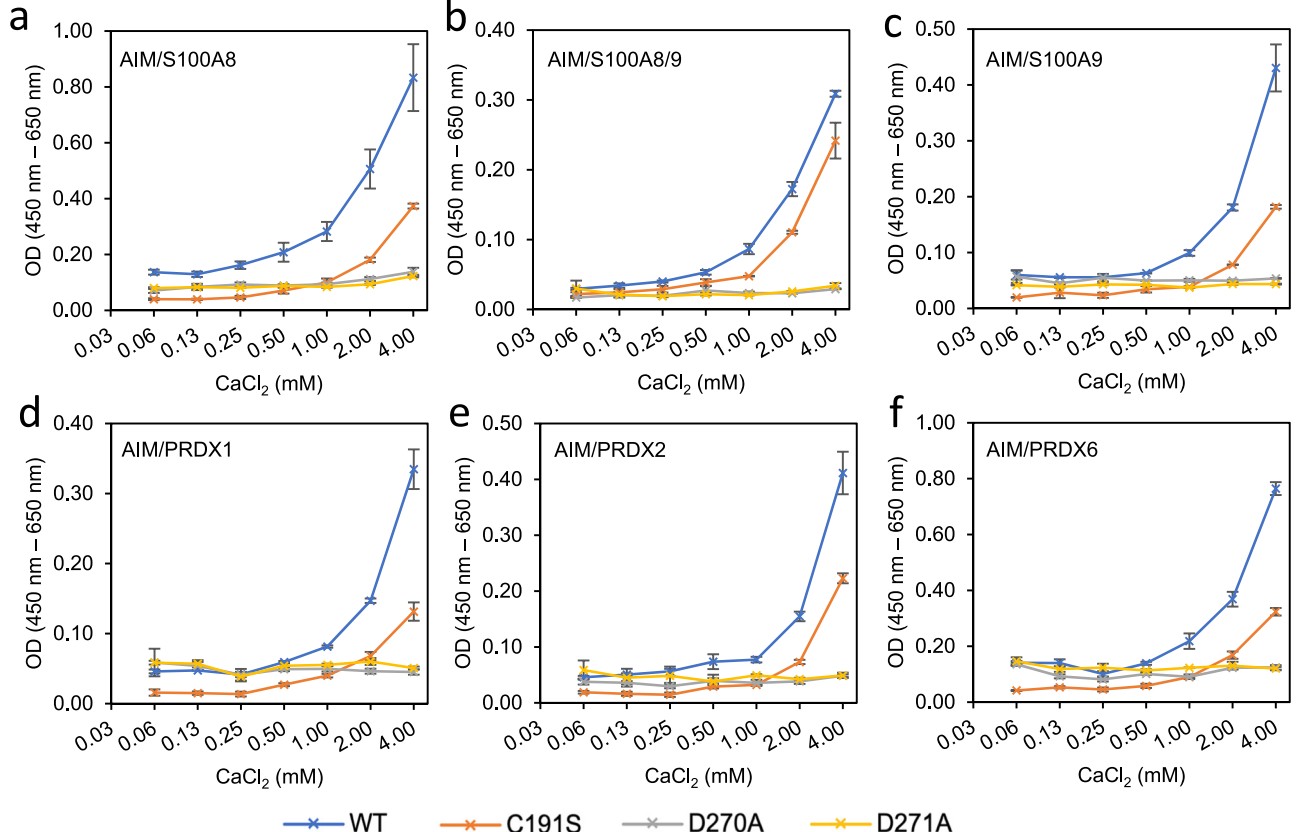

**Fig. 4 | Calcium concentration dependency of AIM and DAMPs binding.**
**a–f** Binding of WT-/mutated-AIM to DAMPs at different calcium ion concentrations in an ELISA-based assay. The DAMPs are (**a**) S100A8. **b** S100A8/9. **c** S100A9. **d** PRDX1. **e** PRDX2. **f** PRDX6. Error bars indicate the s.e.m. for three technical replicates. Data are representative of two biological replicates.

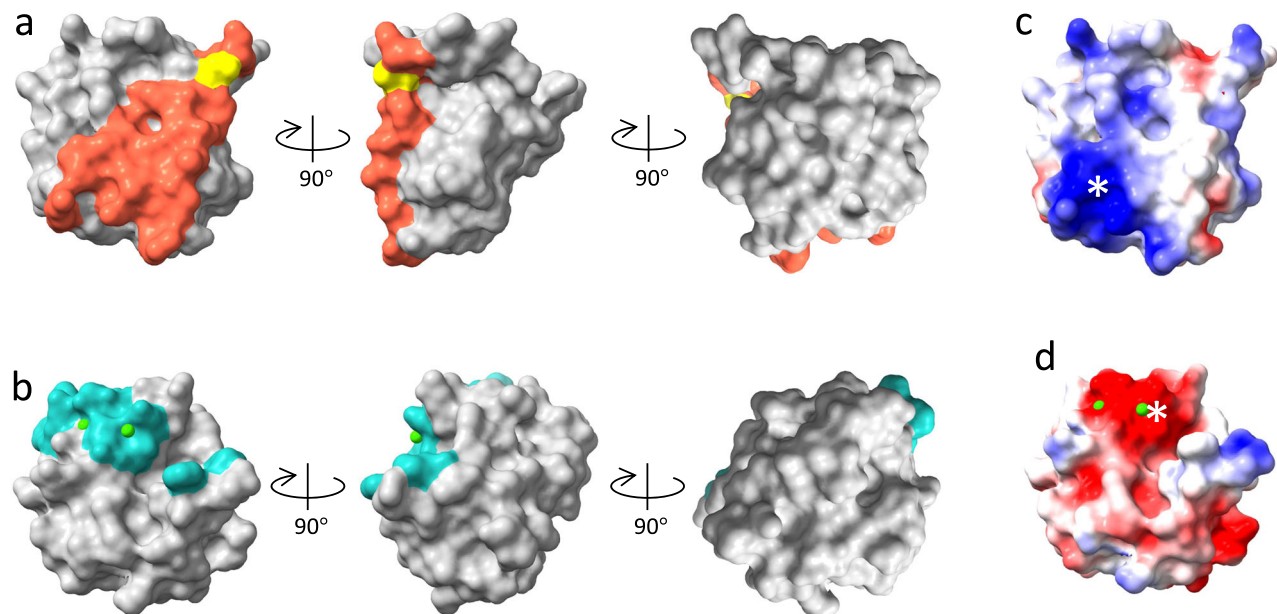

**Fig. 5 | AIM SRCR domain surfaces in the recognition of IgM.** SRCR2 (**a**) and SRCR3 (**b**) with the same viewing angle with the residues at binding interfaces highlighted in red with Cys191 in yellow (SRCR2) and cyan (SRCR3). **c**, **d** Electrostatic surfaces of the two SRCR domains with clusters of charged residues at the binding interfaces indicated by white asterisks.

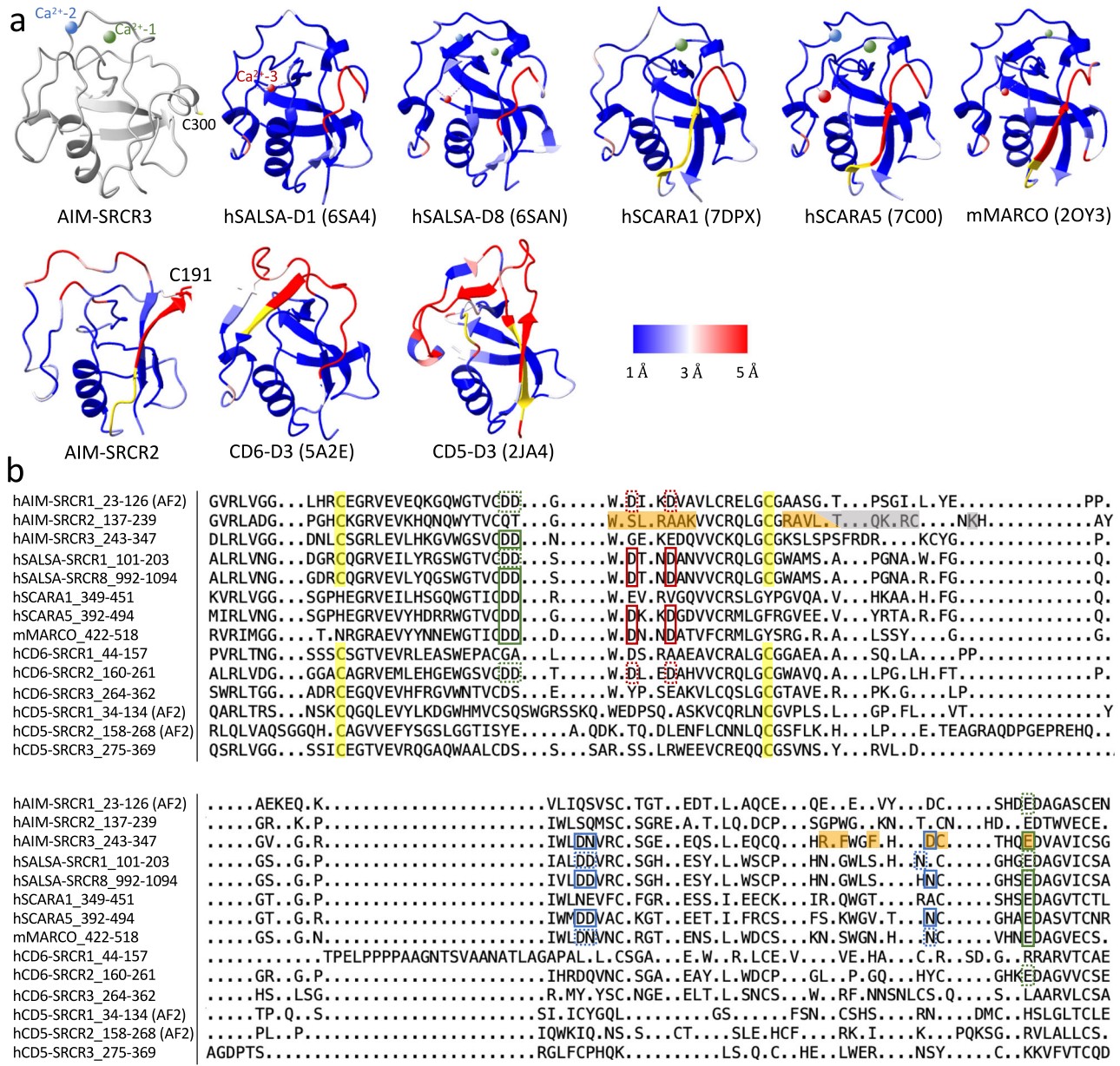

**Fig. 6 | Structural alignment for domains in SRCR super family. a** calculated Ribbon models for various SRCR domains aligned and coloured by Calpha-Calpha distance against AIM SRCR3, with short distance (blue) and long distance (red) according to legend, non-matching residues without a distance calculation in yellow, and AIM SRCR3 structure in grey. The ion binding sites 1, 2 and 3 are coloured in green, blue and red. The larger atoms are calcium, the smaller atoms are magnesium. **b** Structure-based sequence alignment of various SRCR domains by Chimera. hAIM-SRCR1, AlphaFold; hAIM-SRCR2-3, pdb id 8R84; hSALSA-SRCR1, pdb id 6SA4; hSALSA-SRCR8, pdb id 6SAN; hSCARA1, pdb id 7DPX; hSCARA5, pdb id

7C00; mMARCO, pdb id 2OY3; hCD6, pdb id 5A2E; hCD5-SRCR1–2, AlphaFold; hCD5-SRCR3, pdb id 2JA4. The absent C1-C4 disulfide bond in group A SRCR domains (SCARA1, SCARA5 and MARCO) are highlighted in yellow in group B SRCR domains (AIM, CD6, CD5, SALSA). The residues at the binding interface of AIM/IgM are highlighted in orange (AIM/J interface) and grey (AIM/Fcμ interface) in SRCR2 and SRCR3 domains. Three ion binding sites for the SRCR domains are highlighted in green, blue and red boxes. The sites confirmed by structural studies are solid boxes, and the potential ion binding sites where all metal ligands are conserved are in dotted boxes.

results confirm that some key interactions used by AIM for binding to IgM are also used in binding to DAMPs, including both the free cysteine on SRCR2 and the metal ion binding site on SRCR3.

### Relation of AIM SRCR domains to the SRCR super-family
SRCR2 and SRCR3 in AIM engage IgM using interactions that map to distinct parts of the same face of the domain as shown in Fig. 5a, b, where the two aligned domains are viewed in the same orientation (the same view as in Fig. 2b). Additionally, this face shows different clusters of charged residues. In SRCR2, a cluster of positive residues (Fig. 5c) is part of the binding surface for IgM (Fig. 5a) and is complementary to

negative charge on the J-chain; the pair of salt bridges shown in Fig. 3c that are formed in the complex reflects this complementarity. In SRCR3, a cluster of negative residues (Fig. 5d) surrounds the Ca²⁺ sites including Ca²⁺ site 1 that interacts with the J-chain on IgM.

To investigate the similarity and differences among SRCR domains, we examined pairwise structural alignments of AIM-SRCR2 and several other high-resolution SRCR-SF domain X-ray structures against AIM-SRCR3 and calculated Cα-Cα distances between matching residues (Fig. 6a). The associated structure-based sequence alignments are in Fig. 6b. The overall fold of the SRCR domains, whether group A (SCARA1[33], SCARA5[34] and MARCO[26], which lack the C1-C4

disulfide bond highlighted in yellow in Fig. 6) or group B (AIM, CD6[35], CD5[36], SALSA[25]), are all very similar (blue colour in Fig. 6a indicating short Cα-Cα distances), as noted in previous structure comparisons[25]. In the comparison to SRCR3, the largest difference across the SRCR domains analysed is in the ~20 amino acid segment that precedes the free cysteine in SRCR2 (C191) and SRCR3 (C300), which may indicate additional functions for this region and the cysteine in SRCR3.

The structures also show the three metal ion binding sites (Ca²⁺ or Mg²⁺) that are observed in SRCR domains[25]. These include sites 1 and 2 that are present in AIM SRCR3. Ca²⁺ site 1 that bridges SRCR3-AIM and IgM at the J chain is observed in both group A and B proteins in the comparison (first row in Fig. 6a, green boxes in Fig. 6b). Amongst these structures, site 1 has been shown to be essential for calcium-dependent ligand recognition by SCARA1[33], SCARA5[34], SALSA-SRCR8[25] and MARCO[26]. The second Ca²⁺ site is also observed in SALSA and SCARA5 (blue boxes in Fig. 6b) but is likely conserved in others and likely makes some contribution to ligand binding[34]. A third ion site (site 3) described for SALSA, SCARA5 and MARCO is not surface-exposed and has been suggested to contribute to structural integrity for those domains[25] (red solid boxes in Fig. 6b) rather than ligand recognition. Interestingly, the protein fold around the ion binding sites is similar in SRCR domains whether one, two or three ion sites is observed. Though absent in AIM SRCR2, Ca²⁺ site 1 is also likely to be present in AIM SRCR1 based on sequence conservation.

In contrast, several SRCR domains that have neither ions bound nor conserved ion binding residues (structures in the second row in Fig. 6a) have very different structures around the corresponding regions such as CD6 SRCR3 (or CD6 D3), the membrane proximal domain of the three consecutive SRCR domains visualised in the X-ray structure of the CD6 ectodomain[35]. Though there is no Ca²⁺ binding site or free cysteine in CD6-D3, mutagenesis[35] indicates that CD6 binds its ligand CD166 with charge complementary interactions on the same face of the domain used by AIM to bind IgM (Supplementary Fig. 9c).

## Discussion

The human Fcμ/J/AIM complex shows the binding of tandem AIM SRCR domains to IgM leading to sequestration and stabilisation of AIM. This is a visualisation of what may be a common mode of recognition by scavenger receptors given the frequent presence of multiple SRCR domains within a single polypeptide, as in group B SRCR-SF receptors[1]. Multiple SRCR domains in a single polypeptide or in oligomers may enable high-avidity interactions with weak binding sites on diverse ligands. The complex shows adaptations of the SRCR scaffold for different binding modes. The individual AIM SRCR binding domains have specialisations in their interaction with IgM, including an unusual cysteine in SRCR2 which makes a disulfide linkage to Fcμ, and a metal ion binding site on SRCR3 that is present in other SRCR-SF domains. As discussed above, a role for the metal binding site in ligand binding by several SRCR domains has previously been demonstrated by mutagenesis based on X-ray crystal structures[34]. In this study, the Fcμ/J/AIM complex shows directly that a calcium ion bridge between AIM and IgM is the structural basis for calcium-dependent binding in the context of the interfaces at Fcμ and J chain.

For other SRCR domains, charge patches[26,37] on the same binding face have been implicated in ligand binding and studied by mutagenesis. In some cases, the negative charge is associated with the residues surrounding the metal binding sites as shown here in AIM SRCR3. Additionally, the positive charge patch on SRCR2 is complementary to its binding interface on IgM/J. Other SRCR domains can have specialised binding modes based on charge complementarity not dependent on a metal ion as with CD6 membrane proximal D3 domain with CD166 on lymphocytes[35].

Though a detailed mechanism for release of AIM from IgM remains to be elucidated, free AIM exposes canonical SRCR binding surfaces for ligand binding and effector functions. Reduced dissociation of AIM from IgM is associated with disease[13,16] and therefore a structural understanding of the AIM/IgM interface may be applicable in designing therapies that promote release. Specific interactions important for AIM association with IgM are also involved in the recognition of DAMPs. More structural information is needed to fully understand how AIM recognises DAMPs and other targets as well as its association with AIM binding partners that work as effectors. An understanding of AIM recognition of IgM and DAMPs will contribute to therapeutic applications of AIM and other SRCR domain-containing proteins in a wide range of diseases.

## Methods

### Preparation of Fcμ/J/AIM complex

Human Fcμ/J associated with AIM was produced as follows using the ExpiCHO Expression System (Thermo Fisher Scientific). ExpiCHO-S cells transfected with pCAGGS-AIM were co-cultured with ExpiCHO-S cells transfected with both pCAGGS-Flag-Fcμ and pCAGGS-J chain-Myc in ExpiCHO Expression Medium. The culture was supplemented with ExpiCHOFeed and ExpiFectamine CHO Enhancer and maintained for 3 days in a 37 °C incubator with a humidified atmosphere of 8% CO₂ in air on an orbital shaker. First, the Fcμ/J complex (regardless of AIM binding) was purified from the culture supernatant using the ANTI-FLAG M2 Affinity Gel (Sigma-Aldrich). The complex was eluted with 0.1 M glycine-HCl (pH 3.5) and neutralized with 1 M Tris-HCl (pH 8.5). The buffer was exchanged to Dulbecco's phosphate-buffered saline (DBPS, nacalai) using Amicon Ultra filter concentrators 50 K (Millipore). Subsequently, it was passed through a mouse anti-human AIM IgG monoclonal antibody (CL7)-immobilized HiTrap NHS-activated HP column (Cytiva) to obtain Fcμ/J/AIM complex. The protein was eluted with 0.1 M glycine-HCl (pH 2.0) and neutralized with 1 M Tris-HCl (pH 8.5). The buffer was exchanged to DPBS, and the protein was concentrated using Amicon Ultra filter concentrators 50 K. Protein concentration was determined by a BCA assay according to the manufacturer's protocol (Pierce).

### Preparation of AIM mutants (WT, D50A, D51A, D270A, D271A, C191S, D311A, N312A, D334A)

Human AIM protein (WT, D50A, D51A, D270A, D271A, C191S, D311A, N312A, D334A) was produced as follows using the ExpiCHO Expression System. ExpiCHO-S cells were transfected with pCAGGS-AIM with each mutation and cultured in ExpiCHO Expression Medium supplemented with ExpiCHOFeed and ExpiFectamine CHO Enhancer for 4 days in a 37 °C incubator with a humidified atmosphere of 8% CO₂ in air on an orbital shaker. The supernatant was passed through a mouse anti-human AIM IgG monoclonal antibody (CL7)-immobilized HiTrap NHS-activated HP column and the protein was eluted with 0.1 M glycine-HCl (pH 2.5) and neutralized with 1 M Tris-HCl (pH 8.5). The eluate was size-fractionated using HiLoad 16/600 Superdex 30 pg (Cytiva) to further eliminate undesired contaminants. The buffer was exchanged to DPBS, and the protein was concentrated using Amicon Ultra filter concentrators 10 K. Protein concentration was determined by a BCA assay according to the manufacturer's protocol.

### Preparation of DAMPs

PRDXs (PRDX1, 2 and 6) with HA tag were prepared as follows. HEK293T cells (ATCC) were transfected with pCAGGS-PRDX-HA using Lipofectamine 2000 (Thermo Fisher Science) and cultured in DMEM supplemented with 10% FBS for 5 days. The PRDX-HA protein was purified from the supernatant using anti-HA affinity matrix (Roche) following the manufacturer's protocol. S100A8, S100A/8 and S100A9 proteins were purchased from R&D Systems (#9876-S8-050, #8226-S8-050 and #9254-S9-050, respectively).

### Fcμ/J/AIM complex cryo-EM grid preparation

1% octyl glucoside (OG) solution was added into Fcμ/J/AIM (1.3 mg/ml) to a final OG concentration of 0.1%. Quantifoil (200 Cu mesh, R2/2) grids were glow discharged (EMITECH K100X) with air (25 mA, 30 s). 4 μl of sample solution was pipetted to the grid in the environmental chamber of a Vitrobot Mark IV (FEI/Thermo) at 4 °C and 95-100% humidity. The grid was blotted for 4 s before plunging into liquid ethane kept at liquid nitrogen temperature.

### Cryo-EM collection

The IgM cryo grids were first screened on Talos Arctica microscope (FEI/Thermo) at 200 kV and the best ones were transferred to a Titan Krios microscope (FEI/Thermo) at 300 kV equipped with a Gatan Imaging Filter (GIF) using EPU software (v 2.11). The slit width of the energy filter was set to 20 eV. 87,467 movies were recorded on a K2 camera in counting mode with a total dose of 52.4 electrons per $Å^2$ fractionated over 40 frames (dose rate 10.48 $e^-/Å^2/s$) with a 0.846 Å pixel size and a defocus range between -1.2 to -3.5 μm.

### Cryo-EM data processing

The workflow of the Cryo-EM data processing is shown in Supplementary Fig. 1. All movies were imported into Relion (v 4.0.0)[38], followed by Relion's own motion correction and CTF estimation (CTFFIND, v 4.1.13)[39]. 3 M particles in total were picked by a trained model in CrYOLO (v 1.8.2)[40] which were then extracted in Relion with a box size of 128 pixels (bin4, pixel size = 3.384 Å). The particles were then subjected to 2D classification in CryoSPARC, with 700k particles selected based on the populations and resolutions of the class averages. Typical 2D class averages are shown in Supplementary Fig. 1b. The selected particles were re-extracted again in Relion with a box size of 256 pixels (bin2, pixel size = 1.692 Å). 3D classification was performed for the re-extracted particles (Supplementary Fig. 1c) and a final 500k particles were selected and extracted with 600-pixel box size without binning. The particles were then auto-refined and Bayesian polished in Relion, followed by a non-uniform refinement in CryoSPARC (v 4.2.1)[41] with CTF refinement. The final map for Fcμ/J/AIM is shown in Supplementary Fig. 1d. The local resolution map is present in Supplementary Fig. 1e, and 3DFSC plot (Supplementary Fig. 1f) shows isotropic resolution at all orientations. The angular distribution is shown in Supplementary Fig. 1g. The half-map FSC at 0.143 is 3.57 Å (FSC plot in Supplementary Fig. 1i).

To further improve the local resolution at the AIM binding interface to IgM, a mask around the bottom of the complex was created (Supplementary Fig. 2a), which was then applied to local refinement in CryoSPARC. The focused-refined map was then postprocessed in deepEMhancer[42] (Supplementary Fig. 2b). The local resolution map is present in Supplementary Fig. 2c and the angular distribution is shown in Supplementary Fig. 2d. The global resolution is 3.60 Å for half-map FSC at 0.143 cut-off (Supplementary Fig. 2f).

### Model building and refinement

The initial atomic coordinate model of AIM was the AlphaFold predicted model downloaded from Uniprot database (O43866, CD5L human). The IgM core region of the cryo-EM model (pdb id 8BPF) was used as the initial model for Fcμ/J. Real space refinement in Phenix (v 1.19.2)[43] was performed before manually fixing the clashes and outliers in Coot (v 0.9.8.92)[44]. Several iterations between auto- and manual-refinements were conducted for optimisation. N-Acetylglucosamine (NAG) molecules on the IgM Fcμ chains at Asn563 and the calcium ion were built and refined in Coot.

### Map and model validation

The map-model FSCs show 3.7 Å and 3.6 Å at 0.5 cut-off for the global refined (Supplementary Fig. 1i) and local refined (Supplementary Fig. 2f) map. Peptide chains were validated in Coot and Phenix and carbohydrates were validated using Privateer[45] in CCP-EM (v 1.6.0). The data table for Cryo-EM data collection, processing, and validation statistics are summarised in Supplementary Table 1. The figures are made with UCSF Chimera (v 1.13.1) and UCSF ChimeraX (v 1.6.1)[46]. The Cα-Cα distance between matching residues in aligned SRCR domains was calculated using the per-residue root mean square deviation (RMSD) tool in ChimeraX (v 1.6.1)[46]. In Supplementary Fig. 9c, CD6-SRCR3 (pdb id 5A2E) residues implicated in CD166 binding were based on Table S2 in Chappell et al.[35].

### In vitro binding assay for AIM and Fcμ/J or DAMPs

96-well ELISA plates were coated with Fcμ/J or DAMPs resolved in Bicarbonate buffer (0.1 M $NaHCO_3/Na_2CO_3$, pH 9.6) at the concentration described below O/N at 4 °C. The concentrations for the coating proteins are, human Fcμ/J, 10 μg/ml; S100A8, A100A8/9, S100A9, PRDX1-HA, PRDX2-HA and PRDX6-HA, 2 μg/ml. After discarding the protein solution, blocking buffer (1% Casein/TBS-T) was added and incubated for 2 h at room temperature. In the meantime, recombinant AIM was diluted at 20 μg/mL in blocking buffer with various concentration of $CaCl_2$ (0. 0.3125, 0.625, 0.125, 0.25, 0.5, 1.0, 2.0, 4.0 mM) in the presence of 100 μM glutathione and incubated for 1 h at room temperature to reduce the free cysteine residue located at SRCR2 (C191). After blocking, the blocking buffer was removed and the AIM solution with various concentration of $CaCl_2$ was added to the well and incubated for 1 h at room temperature. After 4-times wash with TBS-T, biotinylated anti- human AIM antibody (clone #7) was added to the wells and incubated for 1 h at room temperature. After 4-times wash with TBS-T, the Streptavidin-HRP diluted in 0.2% Casein/ TBS was added to the wells and incubated for 1 h at RT. After 4-times wash with TBS-T, TMB was added and incubated at RT for 2–5 min. The reaction was stopped by adding 1 N $H_2SO_4$ and thereby the absorbance at OD 450 - OD 650 nm was analysed using a multiple plate reader. Binding of AIM to Fcμ/J was performed with the same protocol in the presence of 2.0 mM of various divalent cation chlorides ($CaCl_2$, $MgCl_2$, $MnCl_2$, $CoCl_2$, $NiCl_2$, $ZnCl_2$) in triplicate wells in a single experiment.

### Reporting summary

Further information on research design is available in the Nature Portfolio Reporting Summary linked to this article.

## Data availability

The structural data that support the findings of this study have been deposited in the Protein Data Bank and EM Data bank. The global refined map and model has entry number EMD-18993 and PDB-8R83. The local refined map and model has entry number EMD-18994 and PDB-8R84. Source data are provided with this paper. The published structures used in this study are listed below. 6SA4; 6SAN; 7DPX; 7C00; 2OY3; 5A2E; 2JA4; 8BPF; 8ADY; 6KXS; 6UE7; 6UEA. Source data are provided with this paper.

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

## Acknowledgements

We thank A. Nans of the Structural Biology Science Technology Platform for advice on data collection and computing; A. Purkiss and P. Walker of the Structural Biology Science Technology Platform; the Scientific Computing Science Technology Platform for computational support; E. Hiramoto of Kyowa Kirin Co., Ltd. for protein preparation. This work was supported by the Francis Crick Institute, which receives its core funding from Cancer Research UK (CC2106 (P.B.R.)), the UK Medical Research Council (CC2106 (P.B.R.)), and the Wellcome Trust (CC2106 (P.B.R.)).

This work is also funded by Japan Agency for Medical Research and Development (AMED-LEAP, grant# 23gm0010006h0005 (T.M.)), the Research Grant by Gutenberg Circle, Strasbourg (T.M.) and Japan Society for the Promotion of Science (Grants-in-Aid for Scientific Research KAKENHI, grant# 21H05122 and 23H02684 (S.A.)).

## Author contributions

Q.C., K.I., A.N., S.A., T.M. and P.B.R. contributed to experimental design, data analysis and manuscript writing. Q.C. performed the cryo-EM imaging and data analysis. K.I. conducted the biochemical experiments. H.M. prepared the protein samples.

## Funding

## Competing interests

The authors declare no competing interests.
