## [Transparent Peer Review file · Nature Communications]

Cryo-EM reveals structural basis for human AIM/CD5L recognition of polymeric immunoglobulin M

Corresponding Author: Dr Peter Rosenthal

Version 0:

Reviewer comments:

Reviewer #1

(Remarks to the Author)

This study reports the molecular structure of the complex between pentameric antibody IgM-Fc and the soluble scavenger protein known as AIM, determined by cryo-EM. The structure is well-determined and the details of the analysis well reported in the Methods and Extended Data Figures 1 and 4. The effective resolution in the best determined regions is about 3.6Å. This interaction between AIM and IgM is intriguing: the role of AIM is to bind molecules at sites of infection or trauma that display so-called “damage-associated molecular patterns” or DAMPs, enhancing their phagocytosis. Binding of AIM to IgM appears to serve to enhance levels of AIM in serum, perhaps transporting it to sites of infection, but little is known about how AIM is released from IgM, and this study admits to shedding no light on the release process. However, this is the first detailed structural analysis of the interaction between AIM and IgM-Fc, and is therefore an important first step and a significant contribution to the field. The authors do however attempt to extrapolate from the interactions seen for AIM with IgM, to interactions of AIM and other scavenger molecules of the same structural superfamily (the scavenger receptor cysteine-rich, or SRCR superfamily) with DAMPs, and here the argument is less persuasive and rather poorly presented.

The binding interface between IgM-Fc and two of AIM's three SRCR domains, SRCR2 and 3 (SRCR1 is apparently disordered) is well illustrated in Figs. 1-3. However, the resolution of at best 3.6Å should be stated clearly in the text (it is only mentioned in the Extended Data figures) so that details of H-bonds for example in Figs. 2 and 3 should be treated with some caution. Furthermore, reference is made in several places to “hydrophobic contact”, not only in the text but also in figure legends 2 and 3. What does this mean? Are the interactions shown by the dashed lines van der Waals contacts? If so, then this is what they should be called.

I am also not clear about the identification of the calcium ion binding site. Surely other cations such as Zn²⁺ could fit the density just as well. Although the experimental evidence for calcium is clear (Fig. 3e), were other ions tested? It is not made clear whether this calcium dependence of AIM binding to IgM is a new discovery, or was known before. Are any other SRCR domain structures (perhaps determined by X-ray crystallography) known to have calcium binding sites? This needs to be clarified and referenced.

The matter of the calcium binding site(s) is very confusingly described. At line 191 on page 6 there is the first mention of “calcium binding sites” in the plural; up to then there was apparently only one. Reference is made to Extended Data Fig. 6, and a second site coloured in blue; this is not explained; what is the evidence for these residues as a calcium binding site? In the legend to Extended Data Fig. 6 three calcium binding sites are mentioned! Also in this legend there is no mention of the boxed residues; I understand why D270, D271 and E339 in SRCR3 might be boxed, since these are ligands for the observed calcium ion, but why D334 in SRCR3 and E232 in SRCR2? At lines 197-200 on page 7, “The second ion binding site in AIM... shares an aspartic acid residue with the calcium at the IgM interface”; which residue is this, and is no second calcium ion observed? Further explanation and/or illustration is required.

I am also not convinced by the statement that the two SRCRs of AIM use “the same face of the protein for binding” (lines 169-170, page 6). They are distinct regions (Figs. 4b & e). I also find the extrapolation to the idea that these “same interfaces” are involved in interactions of AIM and other SRCR family members with DAMPs, is very weak. Certainly there is interesting and convincing data in Fig. 4g-i that the presence of calcium enhances binding for several different DAMPs, but perhaps the absence of calcium destabilises the SRCR domains and causes more extensive conformational changes beyond the immediate calcium binding site(s). This could be checked experimentally very easily. In short, the statement that

“AIM binds to IgM and DAMPs in a similar manner...” (line 181, page 6) depends upon what is meant by “similar”; there is certainly a calcium dependence. As for the disulphide bridge with C191, do all the DAMPs have a free cysteine with which to form a bridge?

In summary, the calcium binding site(s) need to be described more clearly, including reference to any structures of other SRCRs where they have been identified. I also think that the claim that “.....similar binding modes are used universally in SRCR recognition of targets in innate immunity” (lines 47-48 of the Summary) is not supported by the data, and should be toned down, both here and in the final three paragraphs. These issues would need to be addressed in a revision of the manuscript. However, this manuscript presents an important and fascinating structure, which should be published, and it is not necessary to exaggerate these extrapolations from the AIM/IgM-Fc interface to other interactions. Not least among the interesting aspects of this structure is the mechanism of release of AIM from IgM, particularly since a disulphide bond is an essential feature of the interaction.

Minor points:

1. p.2 line 64. “...appears to be hindered...” is rather vague; should it be “prevented”? The structure would suggest that it is prevented, especially if interfaces similar to the AIM/IgM-Fc interface are involved.
2. p.4 lines 123-124. It is stated that the linker region between SRCR1 and SRCR2 is “about twice as long” as that between SRCRs 2 and 3. This is unnecessarily vague – how long is the linker? Why not incorporate the two linker sequences into Fig. 2a?
3. p.5 lines 139-140. The shift in the position of the C μ 3-5B domain by AIM binding is said to lead to “asymmetric behaviour of the Fab”. There will be asymmetry in the structure of subunit 5’s Fc region and perhaps a consequent asymmetry in the disposition of the two Fabs, but this needs to be described more fully – and “behaviour” is the wrong word.
4. p.6 before line 169. Suggest insertion of a new sub-heading here.
5. p.7 line 194. “This calcium ion site...” Which site is this?
6. p.7 line 205. No evidence is provided for any cooperativity in binding of the two domains.
7. p.7 line 210. “While the geometry of binding may be similar...” What does this mean?
8. Fig. 2a legend: need to state what is referred to by the three black arrows.

Reviewer #2

(Remarks to the Author)

The Rosenthal group presents a cryo-EM structure, demonstrating the interaction between the SRCR domains of AIM and Fc μ of IgM. This work offers valuable insights into the structural mechanisms that maintain high AIM levels in the blood, and how AIM’s recognition of DAMPs appears to be hindered when associated with IgM. The manuscript is well-written, the figures are visually appealing, and I propose that the paper be considered for publication after minor revision.

Here are my comments and suggestions for improvement:

1. In Extended Data Fig.1. e, k, the dotted blue line is confusing. The authors might want to consider separating the Map-model FSC and Half-Map FSC into two distinct figures for improved clarity.
2. For Extended Data Fig.1. c, d, h, i, it would be beneficial to include the particle number to further enhance the comprehensibility of the data.
3. When using cryoEM densities in Figures, such as Fig. 1c and Extended Data Fig.4a-g, the inclusion of contour levels would assist readers in better understanding the data.
4. The authors should include the local resolution of the local refinement map to provide more detailed information.
5. The last table needs to adhere to the standard format of Nature papers. Currently, it lacks crucial information such as Resolution (global, Å) FSC 0.5 (unmasked/masked), and Map sharpening methods.
6. My main concern is the functional insights from the structure. I urge the authors to delve deeper into the functional implications of this structure. For instance, what would be the functional consequence if the Ca²⁺ binding site is disrupted? Why is the binding interface between AIM and IgM so critical? Can this binding interface be leveraged to design therapeutic interventions for certain diseases? I would appreciate the authors’ insights on these questions.

Reviewer #3

(Remarks to the Author)

In this manuscript, Chen and colleagues present the cryo-EM structural organization of the human CD5L complex with IgM.

This work is highly relevant, as it represents the first report on the cryo-structure of CD5L, a master regulator of inflammatory responses and, consequently, a protein involved in significant pathological situations. Moreover, the work presented contributes to a better understanding of the interaction between IgM and CD5L, which has been previously shown to be highly relevant for the regulation of IgM activity in the context of autoantibody production under obese conditions by the same group (Arai et al., Cell Rep, 2013). Additionally, the presented studies may impact the field of innate immunity, as CD5L belongs to a structurally conserved family of proteins with overlapping functions in the recognition of PAMPs and DAMPs, as well as in the regulation of immune responses.

The manuscript is well written, the methodology is well described and the results well explained in the context of the available literature. However, a discussion section is missing. While the reported structure is of high relevance, the results presented here offer limited novelty for a Journal such as Nature Communications in terms of the biological significance of the data. A recent report (Oskam et al, PNAS 2023) already described the relevant sites of CD5L-IgM interaction and the competition with IgM Fc receptors (pIgR, Fc R and Fc R), providing context for the significance of CD5L interaction in reducing IgM binding to both pIgR and Fc R. Similarly, binding of CD5L to various alarmins through charge-based interactions shown in Figure 4 was previously reported by the same authors in the context of ischemic brain stroke (Maehara et al Cell Rep. 2021). In this regard, it would be interesting to perform competition analyses, and determine the specificity of such interactions.

In conclusion, although the results of this manuscript are of high relevance to the field of SRCR molecules and innate immunity, by providing a structural basis for the CD5L-IgM interaction, they add little novelty in terms of biological activity of CD5L.

Finally I strongly recommend changing the name of AIM for CD5L (CD5 molecule like), as this is the official nomenclature for this protein, as listed in HGNC approved gene nomenclature (HGNC:1690)

Version 1:

Reviewer comments:

Reviewer #1

(Remarks to the Author)

I am satisfied that the authors have addressed all of my original concerns in this substantially revised manuscript, and recommend publication.

Reviewer #2

(Remarks to the Author)

The authors have fully addressed my concerns and I have no more questions. Congrats to all authors.

RESPONSE TO REVIEWER COMMENTS

We thank all the Reviewers for their detailed comments on the ms. We have revised the text as outlined in responses to each Reviewer report below. We have made the following changes to the **Figures**: Figure 3e, AIM binding to IgM, the panel is new data for the same type of experiment we reported previously, but now including 6 AIM mutants. An additional 3 residues are mutated at the second calcium binding site. Figure 3d now includes a second calcium site which does not contact AIM directly. We have also changed the residue numbers in the text and in Figure 3 for the J-chain, which in our original submission was numbered from the signal peptide, not from the mature polypeptide, and we have corrected that. We have updated our deposited models to reflect the renumbering and addition of calcium site 2. We provide the new validation reports. New Figure 5 is a better treatment of material that was originally part of Figure 4 (4a-f). Figure 6 is based on original Supplementary Figure 6 with additions. We have also added two supplementary figures, a new density figure (S3), and a new figure (S7) comparing binding surfaces on AIM SRCR domains and CD6.

Reviewer #1 (Remarks to the Author):

This study reports the molecular structure of the complex between pentameric antibody IgM-Fc and the soluble scavenger protein known as AIM, determined by cryo-EM. The structure is well-determined and the details of the analysis well reported in the Methods and Extended Data Figures 1 and 4. The effective resolution in the best determined regions is about 3.6Å. This interaction between AIM and IgM is intriguing: the role of AIM is to bind molecules at sites of infection or trauma that display so-called “damage-associated molecular patterns” or DAMPs, enhancing their phagocytosis. Binding of AIM to IgM appears to serve to enhance levels of AIM in serum, perhaps transporting it to sites of infection, but little is known about how AIM is released from IgM, and this study admits to shedding no light on the release process. However, this is the first detailed structural analysis of the interaction between AIM and IgM-Fc, and is therefore an important first step and a significant contribution to the field. The authors do however attempt to extrapolate from the interactions seen for AIM with IgM, to interactions of AIM and other scavenger molecules of the same structural superfamily (the scavenger receptor cysteine-rich, or SRCR superfamily) with DAMPs, and here the argument is less persuasive and rather poorly presented.

We thank the Reviewer for both the positive and critical comments and have revised the ms accordingly. We have made our assertions about the relationship between AIM and the SRCR superfamily more precise and hopefully clearer in a new Results section “Relation of AIM SRCR domains to the SRCR super-family” and in the Discussion section that now follows it.

The binding interface between IgM-Fc and two of AIM’s three SRCR domains, SRCR2 and 3 (SRCR1 is apparently disordered) is well illustrated in Figs. 1-3. However, the resolution of at best 3.6Å should be stated clearly in the text (it is only mentioned in the Extended Data figures) so that details of H-bonds for example in Figs. 2 and 3 should be treated with some caution. Furthermore, reference is made in several places to “hydrophobic

contact”, not only in the text but also in figure legends 2 and 3. What does this mean? Are the interactions shown by the dashed lines van der Waals contacts? If so, then this is what they should be called.

P. 3, lines 92-95: We have added the following comment about resolution to the main text : *“The global map resolution is 3.6 Å, with local resolution at the interface predominantly ranging from 3 Å to 3.5 Å. This enabled us to identify density associated with side chains (Supplementary Fig. 2, Supplementary Fig. 3) and propose interactions.”* We have also modified the text to be clearer about what can be confidently interpreted from our experimental data. We replaced all mentions of ‘hydrophobic contact’ with ‘van der Waals contact’ in the main text and Figure legends and agree fully with the Reviewer.

I am also not clear about the identification of the calcium ion binding site. Surely other cations such as Zn²⁺ could fit the density just as well. Although the experimental evidence for calcium is clear (Fig. 3e), were other ions tested? It is not made clear whether this calcium dependence of AIM binding to IgM is a new discovery, or was known before. Are any other SRCR domain structures (perhaps determined by X-ray crystallography) known to have calcium binding sites? This needs to be clarified and referenced.

We cannot be sure from the density alone about the identity of the ions. We have now included an additional experiment in new Supplementary Figure 3e assaying AIM binding using other ions. This supports our interpretation of the density as Ca²⁺.

We have modified the text as follows (Page 4, 123-128): *“There are map densities for two ions at the SRCR3 interface with J chain (Supplementary Fig. 3a-d) at canonical SRCR-SF metal ion binding sites^{25, 26}. The ions are likely to be calcium ions (Ca²⁺) because Ca²⁺ is required for AIM/IgM binding, and binding is reduced when other divalent cations such Mg²⁺ and Zn²⁺ are present instead of calcium (Supplementary Fig. 3e). Density corresponding to metal ions was not identified on SRCR2.”*

The role of calcium was not previously described for AIM except a report that calcium is needed for the ligation of AIM and LPS (Martinez, V.G. et al., Cell Mol. Immunol. 11: 343-354, 2014).

Prior to this study, the structures of SRCR’s have come from X-ray, including the ones we use for structural comparison in original Supplementary Figure 6 (revised text main Figure 6). These include three previously described sites that were found to bind calcium or magnesium, including the locations identified in our complex. These are discussed in the section “Relation of AIM SRCR domains to the SRCR super-family” and in the Discussion.

The matter of the calcium binding site(s) is very confusingly described. At line 191 on page 6 there is the first mention of “calcium binding sites” in the plural; up to then there was apparently only one. Reference is made to Extended Data Fig. 6, and a second site coloured in blue; this is not explained; what is the evidence for these residues as a calcium binding site? In the legend to Extended Data Fig. 6 three calcium binding sites are mentioned! Also in this legend there is no mention of the boxed residues; I understand

why D270, D271 and E339 in SRCR3 might be boxed, since these are ligands for the observed calcium ion, but why D334 in SRCR3 and E232 in SRCR2? At lines 197-200 on page 7, “The second ion binding site in AIM.... shares an aspartic acid residue with the calcium at the IgM interface”; which residue is this, and is no second calcium ion observed? Further explanation and/or illustration is required.

We are sorry for the lack of clarity on this key point. We have revised the ms to address this and also included some additional experiments. Across the SRCR families there are 3 potential metal ion binding sites. We interpret our maps as showing two ions which explains our labelling of Extended Data Figure 6 (now main text Figure 6).

Because the second calcium site is not directly involved in the interface, we did not include this in the original Figure 3 and in the main text we said it was not in position to interact directly with IgM. We have now performed experiments indicating that mutation of ligands of the second site (D311A, N312A, D334A) also reduces AIM binding to IgM, likely through their effect on the first calcium binding site. The second metal ion is now included in Figure 3d. We hope the new sections (p.6, lines 170-185) of the ms address the concerns.

I am also not convinced by the statement that the two SRCRs of AIM use “the same face of the protein for binding” (lines 169-170, page 6). They are distinct regions (Figs. 4b & e). I also find the extrapolation to the idea that these “same interfaces” are involved in interactions of AIM and other SRCR family members with DAMPs, is very weak. Certainly there is interesting and convincing data in Fig. 4g-i that the presence of calcium enhances binding for several different DAMPs, but perhaps the absence of calcium destabilises the SRCR domains and causes more extensive conformational changes beyond the immediate calcium binding site(s). This could be checked experimentally very easily. In short, the statement that “AIM binds to IgM and DAMPs in a similar manner...” (line 181, page 6) depends upon what is meant by “similar”; there is certainly a calcium dependence. As for the disulphide bridge with C191, do all the DAMPs have a free cysteine with which to form a bridge?

We have extensively changed our discussion of the SRCR family in the new section “Relation of AIM SRCR domains to the SRCR super-family” (p.7-8, lines 203-242).

The DAMPs used in our experiments have free cysteines. Generally, free cysteines may become available on DAMPs due to structural changes from damage during release from cytosol as DAMPs.

In summary, the calcium binding site(s) need to be described more clearly, including reference to any structures of other SRCRs where they have been identified. I also think that the claim that “.....similar binding modes are used universally in SRCR recognition of targets in innate immunity” (lines 47-48 of the Summary) is not supported by the data, and should be toned down, both here and in the final three paragraphs. These issues would need to be addressed in a revision of the manuscript. However, this manuscript presents an important and fascinating structure, which should be published, and it is not necessary to exaggerate these extrapolations from the AIM/IgM-Fc interface to other

interactions. Not least among the interesting aspects of this structure is the mechanism of release of AIM from IgM, particularly since a disulphide bond is an essential feature of the interaction.

We have revised the ms accordingly and thank the Reviewer for the suggestions.

Minor points:

1. p.2 line 64. "...appears to be hindered..." is rather vague; should it be "prevented"? The structure would suggest that it is prevented, especially if interfaces similar to the AIM/IgM-Fc interface are involved.

p.3, line 67, we have replaced "is hindered" by "is prevented".

2. p.4 lines 123-124. It is stated that the linker region between SRCR1 and SRCR2 is "about twice as long" as that between SRCRs 2 and 3. This is unnecessarily vague – how long is the linker? Why not incorporate the two linker sequences into Fig. 2a?

We now say (p.5, line 130): *"The linker between SRCR2 and SRCR3 is short, containing 5 residues (C238-F242, Fig. 2c)..."* (p.5, line 135): *"These interactions contribute to the limited flexibility of the linkage between SRCR2 and SRCR3 in contrast to the flexible linker between SRCR1 and SRCR2 domains, which is twice as long, 10 residues, as shown in Fig. 2a."* Fig. 2a legend now says: *"The sequences of linkers between SRCR1 and SRCR2 and between SRCR2 and SRCR3 are underlined in purple."*

3. p.5 lines 139-140. The shift in the position of the C μ 3-5B domain by AIM binding is said to lead to "asymmetric behaviour of the Fab". There will be asymmetry in the structure of subunit 5's Fc region and perhaps a consequent asymmetry in the disposition of the two Fabs, but this needs to be described more fully – and "behaviour" is the wrong word.

p.5, line 154, we have replaced 'behaviour' by 'conformation' and described this more fully.

4. p.6 before line 169. Suggest insertion of a new sub-heading here.

We have added a new sub-heading at p.7, line 203: *"Relation of AIM SRCR domains to the SRCR super-family"*.

5. p.7 line 194. "This calcium ion site..." Which site is this?

This was meant to refer to both the "green" and "blue" sites in Extended Data Figure 6. However, the paragraphs about the two calcium sites have been rewritten.

6. p.7 line 205. No evidence is provided for any cooperativity in binding of the two domains.

"cooperative" is now removed.

7. p.7 line 210. "While the geometry of binding may be similar..." What does this mean?

We have extensively re-written these sections.

8. Fig. 2a legend: need to state what is referred to by the three black arrows.

Figure 2a legend now says of the modified figure: *“The three green arrows indicate the residues of calcium binding site 1 in SRCR3 which are also conserved in SRCR1. The three blue arrows point at the calcium binding site 2 residues in SRCR3. The sequences of linkers between SRCR1 and SRCR2 and between SRCR2 and SRCR3 are underlined in purple.”*

Reviewer #2 (Remarks to the Author):

The Rosenthal group presents a cryo-EM structure, demonstrating the interaction between the SRCR domains of AIM and Fc μ of IgM. This work offers valuable insights into the structural mechanisms that maintain high AIM levels in the blood, and how AIM's recognition of DAMPs appears to be hindered when associated with IgM. The manuscript is well-written, the figures are visually appealing, and I propose that the paper be considered for publication after minor revision.

Here are my comments and suggestions for improvement:

1. In Extended Data Fig.1. e, k, the dotted blue line is confusing. The authors might want to consider separating the Map-model FSC and Half-Map FSC into two distinct figures for improved clarity.

We agree that clarity needs to be improved. We have modified the plots (Supplementary Fig. 1e,l) by addition of labels where the two FSC curves cross the commonly reported resolution thresholds.

2. For Extended Data Fig.1. c, d, h, i, it would be beneficial to include the particle number to further enhance the comprehensibility of the data.

We have added particle numbers to the figure panels as suggested.

3. When using cryoEM densities in Figures, such as Fig. 1c and Extended Data Fig.4a-g, the inclusion of contour levels would assist readers in better understanding the data.

We have added the map contour value and the map sigma level for 1c and Supplementary Fig. 2 (previously Extended Data Fig.4) as well as the newly added density figure, Supplementary Fig. 3. We note that sigma levels can depend on box size of the single particle calculation, sharpening, and other parameters, but agree that these are needed additions.

4. The authors should include the local resolution of the local refinement map to provide more detailed information.

We have added a map of local resolution of the local refinement map as Supplementary Figure 1j.

5. The last table needs to adhere to the standard format of Nature papers. Currently, it lacks crucial information such as Resolution (global, Å) FSC 0.5 (unmasked/masked), and Map sharpening methods.

We have modified the table as requested.

6. My main concern is the functional insights from the structure. I urge the authors to delve deeper into the functional implications of this structure. For instance, what would be the functional consequence if the Ca²⁺ binding site is disrupted? Why is the binding interface between AIM and IgM so critical? Can this binding interface be leveraged to design therapeutic interventions for certain diseases? I would appreciate the authors' insights on these questions.

We have now added the following:

p. 9, lines 268-277: *“Though a detailed mechanism for release of AIM from IgM remains to be elucidated, free AIM exposes canonical SRCR binding surfaces for ligand binding and effector functions. Reduced dissociation of AIM from IgM is associated with disease^{13,16} and therefore a structural understanding of the AIM/IgM interface may be applicable in designing therapies that promote release. Specific interactions important for AIM association with IgM are also involved in the recognition of DAMPs. More structural information is needed to fully understand how AIM recognises DAMPs and other targets as well as its association with AIM binding partners that work as effectors. An understanding of AIM recognition of IgM and DAMPs will contribute to therapeutic applications of AIM and other SRCR domain-containing proteins in a wide range of diseases.”*

Reviewer #3 (Remarks to the Author):

In this manuscript, Chen and colleagues present the cryo-EM structural organization of the human CD5L complex with IgM. This work is highly relevant, as it represents the first report on the cryo-structure of CD5L, a master regulator of inflammatory responses and, consequently, a protein involved in significant pathological situations. Moreover, the work presented contributes to a better understanding of the interaction between IgM and CD5L, which has been previously shown to be highly relevant for the regulation of IgM activity in the context of autoantibody production under obese conditions by the same group (Arai et al., Cell Rep, 2013). Additionally, the presented studies may impact the field of innate immunity, as CD5L belongs to a structurally conserved family of proteins with overlapping functions in the recognition of PAMPs and DAMPs, as well as in the regulation of immune responses.

The manuscript is well written, the methodology is well described and the results well explained in the context of the available literature. However, a discussion section is missing. While the reported structure is of high relevance, the results presented here offer limited novelty for a Journal such as Nature Communications in terms of the biological significance of the data. A recent report (Oskam et al, PNAS 2023) already described the relevant sites of CD5L-IgM interaction and the competition with IgM Fc receptors (pIgR, Fc $\alpha\mu$ R and Fc μ R), providing context for the significance of CD5L interaction in reducing IgM binding to both pIgR and Fc $\alpha\mu$ R. Similarly, binding of CD5L to various alarmins through charge-based interactions shown in Figure 4 was previously

reported by the same authors in the context of ischemic brain stroke (Maehara et al Cell Rep. 2021). In this regard, it would be interesting to perform competition analyses, and determine the specificity of such interactions.

In conclusion, although the results of this manuscript are of high relevance to the field of SRCR molecules and innate immunity, by providing a structural basis for the CD5L-IgM interaction, they add little novelty in terms of biological activity of CD5L.

We thank the Reviewer for the positive comments and supporting the relevance and wide interest of our findings. As the Reviewer has identified, the main focus of our paper is the structure of the complex. We feel the structural data are an important advance in interpreting past experiments and for designing new ones. We expect the cryo-EM structure to be particularly important to understanding data in Oskem et al., a paper which we have cited in the ms. Furthermore, there are many interesting structures of SRCR domains but, to our knowledge, our structure is the first complex of an SRCR domain with its protein ligands and therefore is novel and important to a wide range of biological activities.

We have added a new section “Relation of AIM SRCR domains to the SRCR super-family” and Discussion section which together summarises the importance of our findings and which we hope addresses concerns raised by the Reviewer.

The ms also presents binding assays to show the specific importance of the calcium binding sites and cysteine in binding IgM and DAMPs which complement the structural data. Regarding the Reviewer’s mention of Figure 4 in Maehara et al Cell Rep. 2021, we say in the manuscript (lines 191-194) “*we performed binding assays for six DAMPs previously shown to be bound by AIM³ (Maehara et al., Cell Rep, 2021), with WT or mutated AIM (Fig. 4a-f).*” To be clear, in our new ms we are reporting new experiments that test interactions as a function of calcium and with mutations of residues that we have identified to bind the calcium at the interface between SRCR3 and IgM. Regarding competitive assays, our previous paper (Maehara et al., Cell Rep, 2021) shows IgM -Fc pentamer inhibition of AIM binding to DAMPs . Please refer to Figures 5F and G of that paper.

Finally I strongly recommend changing the name of AIM for CD5L (CD5 molecule like), as this is the official nomenclature for this protein, as listed in HGNC approved gene nomenclature (HGNC:1690)

In the ms we are clear to say that the molecule has both names, AIM and CD5L.

RESPONSE TO REVIEWER COMMENTS

Reviewer #1 (Remarks to the Author):

I am satisfied that the authors have addressed all of my original concerns in this substantially revised manuscript, and recommend publication.

Reviewer #2 (Remarks to the Author):

The authors have fully addressed my concerns and I have no more questions. Congrats to all authors.

We thank the reviewers for their final positive comments.